Subject Areas:
e-science

Keywords:
quota, debiasing, algorithmic fairness, intersectionality

Author for correspondence:
I. Smirnov
e-mail: ibsmirnov@hse.ru

# Quota-based debiasing can decrease representation of the most under-represented groups

I. Smirnov[1,2], F. Lemmerich[3] and M. Strohmaier[1,4]

[1]Department for Society, Technology and Human Factors and Department of Computer Science, RWTH Aachen University, Aachen, Nordrhein-Westfalen, Germany
[2]Institute of Education, HSE University, Moskva, Moskva, Russian Federation
[3]Faculty of Computer Science and Mathematics, University of Passau, Passau, Bayern, Germany
[4]Department of Computational Social Science, GESIS—Leibniz Institute for the Social Sciences, Cologne, Germany

 IS, 0000-0002-8347-6703

Many important decisions in societies such as school admissions, hiring or elections are based on the selection of top-ranking individuals from a larger pool of candidates. This process is often subject to biases, which typically manifest as an under-representation of certain groups among the selected or accepted individuals. The most common approach to this issue is debiasing, for example, via the introduction of quotas that ensure a proportional representation of groups with respect to a certain, often binary attribute. This, however, has the potential to induce changes in representation with respect to other attributes. For the case of two correlated binary attributes, we show that quota-based debiasing based on a single attribute can worsen the representation of the most under-represented intersectional groups and decrease the overall fairness of selection. Our results demonstrate the importance of including *all* relevant attributes in debiasing procedures and that more efforts need to be put into eliminating the root causes of inequalities as purely numerical solutions such as quota-based debiasing might lead to unintended consequences.

## 1. Introduction

Selection of top-ranked individuals from a larger pool of candidates is a ubiquitous mechanism for decision-making. In many countries, school admission is determined by the selection of top graduates based on their test scores. Elections are, in general, a selection of top candidates based on the number of

votes they get. Hiring and promotion are essentially processes of choosing top individuals from a limited pool of candidates based on an implicit ranking of their skills.

Such processes are known to be affected by biases. For example, hiring decisions have been found to be biased with respect to gender [1,2] and ethnicity [3,4]. Given the crucial role that selection processes play in shaping our everyday life and their potentially high-stakes consequences, eliminating—or at least limiting—undesirable biases is essential.

The most common solution is to introduce quotas that ensure the proportional representation of groups with respect to a certain, often binary attribute. Examples include, among many others, quotas for women on corporate boards [5], ethnic quotas in elections [6] and quotas based on the state of origin in university admissions [7]. While successfully eliminating under-representation with respect to one attribute, quotas typically ignore changes in the representation with respect to other attributes. This can lead to unintended consequences and can even decrease the representation of already under-represented groups. For instance, it was found that the introduction of minority quotas in elections—while increasing the representation of minority women—could simultaneously lead to lower representation levels of women in the majority even though this group was already under-represented [8].

## 2. The debiasing paradox

We define the *debiasing paradox* to describe paradoxical situations in which interventions that reduce bias for groups defined by a property can further decrease the representation of an already under-represented or even the most under-represented subgroup. This paradox occurs when other potentially sensitive—but invisible or ignored—attributes are correlated with the attribute that is used for debiasing, which can happen quite naturally in real-world settings. One example is the pay gap between women and men that could partially be explained by the wage penalty for mothers [9]. In this case, two attributes—'being a woman' and 'taking care of children'—are correlated and both could have negative effects on salary. Debiasing on the first attribute might lead to unintended side effects for some minority groups, i.e. women who are not taking care of children or men who do. The debiasing paradox in its weaker form was previously observed empirically, i.e. it was shown that quotas could decrease representation of some under-represented subgroups [8]. However, it remained unclear if the debiasing paradox could occur in its stronger form, that is, if the introduction of quotas could decrease the representation of the most under-represented intersectional group, and how overall fairness of ranking would be affected.

## 3. Theoretical model

We present a theoretical model with correlated binary attributes to demonstrate that debiasing can paradoxically worsen the representation of the most disadvantaged group if a second hidden attribute is taken into account. This can, for example, happen if a discrepancy between aggregated and disaggregated data is observed, cf. Simpson's paradox [10].

We consider a world populated by social entities (individuals) that have a certain inherent quality $q$ for a task, such that $q \sim N(0, 1)$ is normally distributed. The entities have one attribute (e.g. *colour:* green or orange) that is visible to a public, and another attribute (e.g. *shape:* stars or circles) that is invisible or ignored. Both attributes (colour and shape) are correlated with each other. For simplicity, we assume that there are equal numbers of stars and circles ($N$) as well as equal numbers of green and orange entities. That is, there are $f * N$ green circles and $f * N$ orange stars, where $0 < f < 0.5$.

In this setting, we now consider biases in the perception of this quality. We assume that instead of the real quality $q$ the selection of top candidates is based on perceived quality $\hat{q}$, where $\hat{q}_i = q_i - d_{\text{colour}} * I_i^{\text{colour}} - d_{\text{shape}} * I_i^{\text{shape}}$, i.e. the perceived quality is lower than the real quality for entities of a particular colour and a particular shape. Here, the indicator function $I_i^{\text{colour}}$ is 1 for green and 0 for orange entities, $I_i^{\text{shape}}$ is 1 for stars and 0 for circles, $d_{\text{colour}}$ and $d_{\text{shape}}$ are fixed biases. We then explore how debiasing on the visible attribute *colour* affects the representation of all four different subgroups (green stars, green circles, orange stars and orange circles) of entities.

Since in this setting the real quality $q$ is independent from *shape* and *colour*, each group $g$ would be proportionally represented in an unbiased selection of the top $k\%$ candidates. That is, the chances of an entity from group $g$ to appear in the top $k\%$ would be equal to its share among all entities: $N_g/N_{\text{total}}$, where $N_g$ is the size of group $g$ and $N_{\text{total}}$ is the total number of entities.

Through the introduced bias, selection based on the perceived quality $\hat{q}$, however, can lead to different outcomes. Thus, we can compute a *representation bias $B_g$*, i.e. the under- or over-representation of a certain

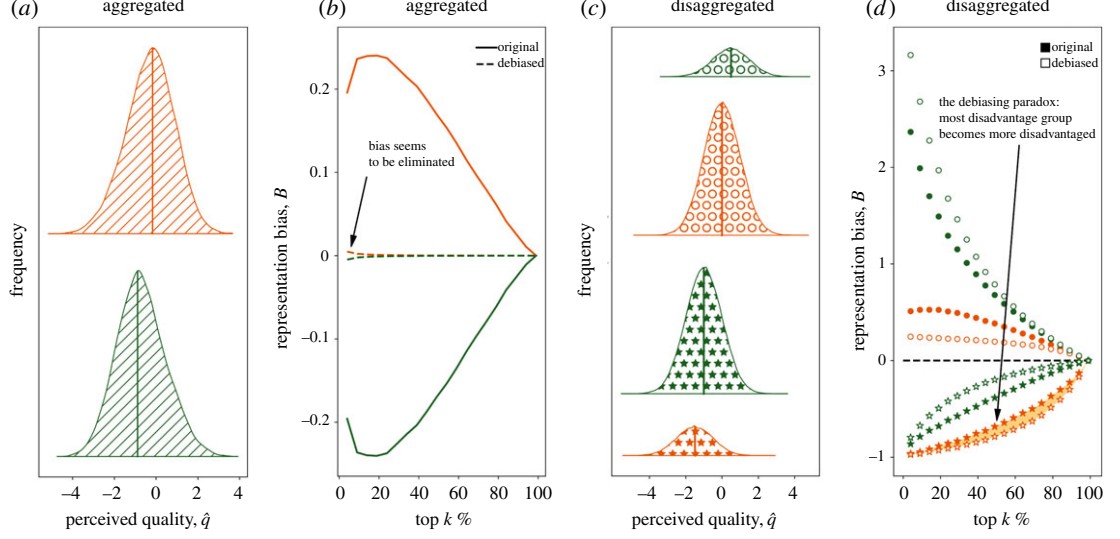

**Figure 1.** The debiasing paradox. If only one attribute (*colour*) is considered then orange entities appear to have an advantage as their average perceived quality is higher (*a*). In fact, being orange is a disadvantage by construction (*c*). In this case, while debiasing on *colour* seems to eliminate colour bias (*b*), it, in fact, affects various subgroups differently (*d*). In particular, it worsens the representation of the already most disadvantaged group of orange stars. We call this effect the debiasing paradox.

group $g$ in biased selections as the relative change in chances for its members to appear in top $k\%$ positions. If the proportion of entities from group $g$ in top $k\%$ positions is higher than $N_g/N_{\text{total}}$ then the group is over-represented, if lower then the group is under-represented. To avoid such misrepresentation and achieve statistical parity, a common solution is to apply quota-based debiasing by allocating a proportional number of positions for each group in the top $k\%$ rankings and filling them with the candidates with highest $\hat{q}$.

Quota-based debiasing is widely used in policy- and decision-making and underpins algorithmic debiasing [11]. In practice often only a single attribute is used for debiasing and other relevant attributes are either unknown or ignored. We demonstrate next that this could lead to unintended consequences.

Figure 1 shows example results for $f = 0.2$, $d_{\text{colour}} = -0.5$ (i.e. greens are perceived as having better quality) and $d_{\text{shape}} = 1.5$ (i.e. stars are perceived as having lower quality). If only colour is considered, then debiasing appears to work as intended by successfully eliminating under-representation: since the perceived quality of green entities is lower than that of the orange entities (figure 1*a*) they are under-represented in top $k\%$ if the selection is blind towards all attributes, but quota-based debiasing on *colour* successfully corrects for that (figure 1*b*). However, the apparent bias against greens contradicts the mechanism generating the data—by construction, a green instance is at an advantage compared to an orange instance with the same shape. The appearance of a bias against green is explained by the fact that greens disproportionately consist of stars (figure 1*c*) and stars are at a larger disadvantage, i.e. the penalty for stars of 1.5 standard deviations is three times larger than the advantage for green entities.

As a consequence, quota-based debiasing would in this particular example improve the position of green circles that are already the most advantaged group and worsen the position of orange stars that are already the most disadvantaged group (figure 1*d*). This illustrates the emergence of a *debiasing paradox* in its strong form, where debiasing can decrease the representation of already under-represented groups.

# 4. Effects on the overall fairness of selection

The goal of a selection process is typically to maximize the average quality $q$ of selected people. If the real quality is uncorrelated with other attributes that only influence the perception, then the unbiased selection of top $k\%$ ranking candidates would achieve this goal as it selects candidates with highest possible $q$. A biased selection is based on perceived quality $\hat{q}$ instead. Thus, the average performance of selected candidates would be typically lower. Thus, we quantify the fairness of a selection $F_k$ as the difference in quality between (i) an average person at top $k\%$ positions of a biased selection and (ii) an

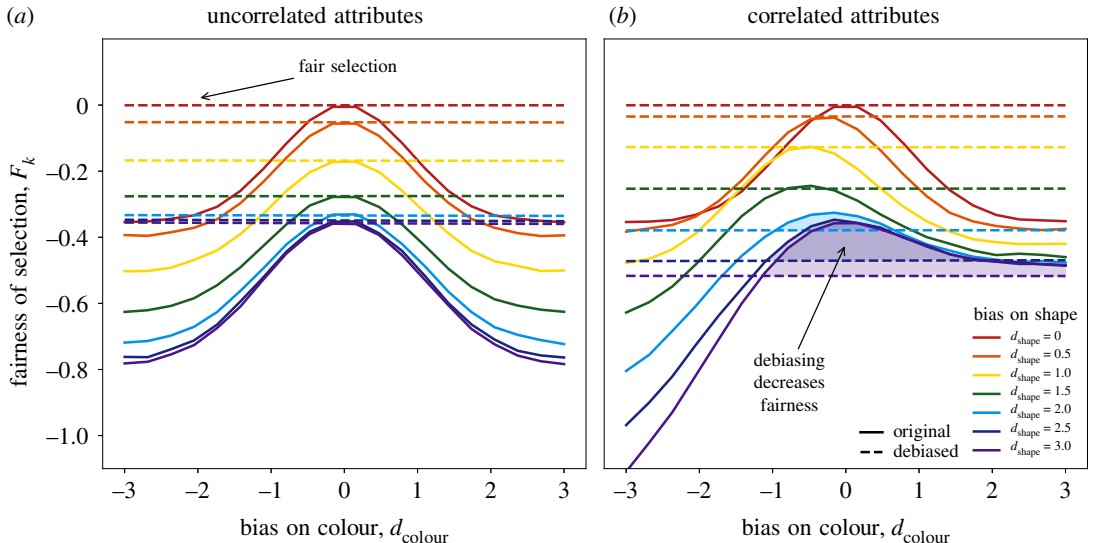

**Figure 2.** The effects of debiasing on the quality of selected candidates for uncorrelated (*a*) and correlated (*b*) attributes. When shape and colour are uncorrelated and there is no bias for shape (red solid line on panel *a*) then debiasing on colour successfully maximizes the quality of selected candidates. If there exists bias for shape then the maximum value is not achieved but quality is still improved. However, if two attributes are correlated then in some cases overall quality *decreases* (shaded regions on panel *b*).

average person at top $k\%$ positions of an unbiased selection. As unbiased selection maximizes the quality of selected participants, the maximum value of $F_k$ is zero.

The debiasing paradox raises the question whether the overall fairness of a selection is improved after debiasing. To answer this question, we explore the changes in the average quality of selected candidates for different values of $d_{colour}$ and $d_{shape}$. While the real quality in real-world data is almost always not observable, we can study this effect with the introduced theoretic model.

Figure 2*a* demonstrates changes in the quality of selected candidates if colour and shape are uncorrelated ($f = 0.5$, $k = 0.2$). If there is no bias for shape (red solid line), then debiasing on colour successfully maximizes the real quality of candidates (red dashed line). In other cases, debiasing would not achieve the maximum value, but would still improve the quality of selected candidates.

Figure 2*b* shows results for the case that colour and shape are correlated ($f = 0.2$, $k = 0.2$). It can be observed that there are cases in which the average quality of selected participants *decreases after debiasing*. One scenario where this would happen is when there is no bias on colour but a large enough bias on shape. This would, for instance, happen when debiasing is based on gender while gendered behaviour and not gender itself is penalized. In this case, introducing quotas would make the selection less fair if the penalty is large enough.

## 5. Discussion

Quota-based debiasing is an effective way to remove bias with respect to a single binary attribute. Our work demonstrates the potential negative side effects of quotas on subgroups. These effects can appear in situations with incomplete knowledge, i.e. when some relevant attributes are unknown or ignored. Our work takes an action-oriented perspective towards these problems, by highlighting the potential unintended consequences of interventions such as quotas. In particular, we show that quota-based debiasing could worsen the representation of already under-represented groups and decrease the overall fairness of rankings. Our work studies these effects from a statistical point of view, but does not consider potential indirect effects and long-term consequences. For instance, quotas could increase the quality of majority candidates [12], increase the diversity of both majority and minority candidates, and change the definition of a quality candidate [13].

While debiasing—and specifically quota-based debiasing—has been applied for decades, the recent rise of artificial intelligence systems amplifies and compounds this problem. Artificial intelligence systems have been shown to reproduce or even enhance biases from training data [14]. This problem could be particularly salient for intersectional groups. For example, Buolamwini and Gebru compared the accuracy of commercial gender classification systems for the four intersectional subgroups and

have found that all studied classifiers perform worst on darker female faces (20.8–34.7% error rate) while the maximum error rate for lighter-skinned males was just 0.8% [15].

Foulds *et al.* suggested that to address challenges of AI fairness with respect to intersectional groups, the definition of fairness in machine learning and artificial intelligence systems should be informed by the framework of intersectionality [16,17]. Several approaches to measuring algorithmic fairness were proposed to account for the presence of intersectional subgroups [18,19]. Some fair ranking algorithms are specifically designed to address intersectionality [20,21].

Such approaches, however, inevitably require taking many attributes into consideration. The increase in the number of dimensions leads to data sparsity that rapidly becomes an issue as some intersectional subgroups become too small for meaningful debiasing [18]. Even if data allow for considering all intersectional subgroups for a given set of attributes, there is still a possibility that the influence of some hidden attributes could lead to unintended consequences.

Our work raises awareness of the fact that automatically applied quotas to achieve statistical parity of groups might not be a suitable solution, and can in some scenarios even worsen disparate representations of subgroups. It means that quota-based debiasing warrants caution and control for various additional attributes. In some cases, it could be impossible to fix bias for one attribute without introducing a bias for another. This shall encourage further research on alternative domain-specific approaches, see for example [22]. Overall, our work demonstrates that instead of solely relying on *post hoc* fixing of biases via quotas or similar numerical solutions, more efforts should be directed towards eliminating the root causes of inequalities.

Data accessibility. Data and relevant code for this research work are stored in GitHub: https://github.com/ibsmirnov/debiasing and have been archived within the Zenodo repository: https://doi.org/10.5281/zenodo.5167942.
Authors' contributions. I.S., F.L. and M.S. designed research and wrote the paper, I.S. analysed the data.
Competing interests. We declare we have no competing interests.
Funding. I.S. acknowledges support from the Basic Research Program of the National Research University Higher School of Economics.

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
