## [Peer Review File · Royal Society Open Science]

Review History

RSOS-210821.R0 (Original submission)

Review form: Reviewer 1

Is the manuscript scientifically sound in its present form?

Yes

Are the interpretations and conclusions justified by the results?

Yes

Is the language acceptable?

Yes

Do you have any ethical concerns with this paper?

No

Have you any concerns about statistical analyses in this paper?

No

Recommendation?

Accept with minor revision (please list in comments)

Comments to the Author(s)

This paper presents an interesting formal expression of a known difficulty in debiasing as it relates to correlated variables. My only concern is that the paper is too narrow, too limited to the statistical argument, and not conversant enough with related literatures. For instance, there has recently emerged a body of research around "intersectional fairness" and "subgroup fairness," which should have been explored in discussion and might have well shaped or enriched the conclusions of the authors. This type of article also would have benefited from spending more time elucidating previous work, so that readers are better informed about related research and literatures.

A few suggested references:

Foulds, J. R., Islam, R., Keya, K. N., & Pan, S. (2020, April). An intersectional definition of fairness. In 2020 IEEE 36th International Conference on Data Engineering (ICDE) (pp. 1918-1921). IEEE.

Buolamwini, J., & Gebru, T. (2018, January). Gender shades: Intersectional accuracy disparities in commercial gender classification. In Conference on fairness, accountability and transparency (pp. 77-91). PMLR.

Foulds, J. R., Islam, R., Keya, K. N., & Pan, S. (2020). Bayesian Modeling of Intersectional Fairness: The Variance of Bias*. In Proceedings of the 2020 SIAM International Conference on Data Mining (pp. 424-432). Society for Industrial and Applied Mathematics.

Yang, K., Loftus, J. R., & Stoyanovich, J. (2020). Causal intersectionality for fair ranking. arXiv preprint arXiv:2006.08688.

Decision letter (RSOS-210821.R0)

Dear Mr Smirnov

On behalf of the Editors, we are pleased to inform you that your Manuscript RSOS-210821 "Quota-based debiasing can decrease representation of the most underrepresented groups" has been accepted for publication in Royal Society Open Science subject to minor revision in accordance with the referees' reports. Please find the referees' comments along with any feedback from the Editors below my signature.

Please submit your revised manuscript and required files (see below) no later than 7 days from today's (ie 05-Jul-2021) date. Note: the ScholarOne system will 'lock' if submission of the revision is attempted 7 or more days after the deadline. If you do not think you will be able to meet this deadline please contact the editorial office immediately.

on behalf of Professor Mark Girolami (Associate Editor) and Marta Kwiatkowska (Subject Editor)
openscience@royalsociety.org

Associate Editor Comments to Author (Professor Mark Girolami):

Please address the comments from the referee in making your revision.

Reviewer comments to Author:

Reviewer: 1

Comments to the Author(s)

This paper presents an interesting formal expression of a known difficulty in debiasing as it relates to correlated variables. My only concern is that the paper is too narrow, too limited to the statistical argument, and not conversant enough with related literatures. For instance, there has recently emerged a body of research around "intersectional fairness" and "subgroup fairness," which should have been explored in discussion and might have well shaped or enriched the conclusions of the authors. This type of article also would have benefited from spending more time elucidating previous work, so that readers are better informed about related research and literatures.

A few suggested references:

Foulds, J. R., Islam, R., Keya, K. N., & Pan, S. (2020, April). An intersectional definition of fairness. In 2020 IEEE 36th International Conference on Data Engineering (ICDE) (pp. 1918-1921). IEEE.

Buolamwini, J., & Gebru, T. (2018, January). Gender shades: Intersectional accuracy disparities in commercial gender classification. In Conference on fairness, accountability and transparency (pp. 77-91). PMLR.

Foulds, J. R., Islam, R., Keya, K. N., & Pan, S. (2020). Bayesian Modeling of Intersectional Fairness: The Variance of Bias*. In Proceedings of the 2020 SIAM International Conference on Data Mining (pp. 424-432). Society for Industrial and Applied Mathematics.

Yang, K., Loftus, J. R., & Stoyanovich, J. (2020). Causal intersectionality for fair ranking. arXiv preprint [arXiv:2006.08688](https://arxiv.org/abs/2006.08688).

===PREPARING YOUR MANUSCRIPT===

===PREPARING YOUR REVISION IN SCHOLARONE===

Author's Response to Decision Letter for (RSOS-210821.R0)

See Appendix A.

Decision letter (RSOS-210821.R1)

Dear Mr Smirnov,

I am pleased to inform you that your manuscript entitled "Quota-based debiasing can decrease representation of the most underrepresented groups" is now accepted for publication in Royal Society Open Science.

on behalf of Professor Mark Girolami (Associate Editor) and Marta Kwiatkowska (Subject Editor)
openscience@royalsociety.org

Appendix A

Dear Editor(s),

thank you for the comments from the reviewer and for the opportunity to refine our manuscript. As suggested by the reviewer, we have included in our paper the discussion of intersectional fairness in AI and machine learning and its relationship to our work (see p.5). We hope that this results in a manuscript that is now appropriate for publication in Royal Society Open Science.

With kind regards on behalf of the authors
Ivan Smirnov